# Effect of Annealing on the Microstructure, Opto-Electronic and Hydrogen Sensing Properties of $V_2O_5$ Thin Films Deposited by Magnetron Sputtering

**Michał Mazur** *, **Szymon Kiełczawa**  and **Jarosław Domaradzki**

Faculty of Electronics, Photonics and Microsystems, Wrocław University of Science and Technology, Janiszewskiego 11/17, 50-372 Wrocław, Poland
*   Correspondence: michal.mazur@pwr.edu.pl

**Abstract:** This paper reports results of investigations on selected properties of vanadium oxide thin films deposited using gas impulse magnetron sputtering and annealed at temperatures in the range of 423 K to 673 K. Post-process annealing was shown to allow phase transition of as-deposited films from amorphous to nanocrystalline $V_2O_5$ with crystallite sizes in the range of 23 to 27 nm. Simultaneously, annealing resulted in an increase in surface roughness and grain size. Moreover, a decrease in transparency was observed in the visible wavelength range of approximately 50% to 30%, while the resistivity of formed vanadium pentoxide thin films was almost unchanged and was in the order of $10^2$ $\Omega\cdot$cm. Simultaneously, the best optoelectronic performance, testified by evaluated figure of merit parameter, indicated the as-deposited amorphous films. Performed Seebeck coefficient measurements indicated the electron type of electrical conduction of each prepared thin film. Furthermore, gas sensing properties towards diluted hydrogen were investigated for annealed $V_2O_5$ thin films, and it was found that the highest senor response was obtained for a thin film annealed at 673 K and measured at operating temperature of 623 K.

**Keywords:** vanadium oxide; thin film; annealing; microstructure; optical properties; electrical properties; hydrogen gas sensing

## 1. Introduction

Metal oxides with electrical and optical properties suitable for the use in opto-electronics and sensor technology have been studied for many years [1]. The current interest in metal oxides is due to its properties, which are determined by oxygen vacancies [2]. One of the chemical compounds which is recently of great interest is vanadium oxide [3–5]. Vanadium in oxide compounds can exist at various oxidation states, including $V_2O_3$, $V_5O_9$, $VO_2$, and $V_2O_5$ [6]. Most vanadium oxides can undergo a semiconductor–metal transition at a certain temperature followed by a sharp and reversible change in their optical or electrical properties [7]. Vanadium dioxide is known for its ability to react to temperature changes by transforming the monoclinic structure into rutile at a temperature of 341 K [7,8]. As this transition takes place at a temperature close to room temperature, it makes them suitable for optical switches and microbolometers [9]. Vanadium pentoxide is also intensively researched. $V_2O_5$ can also undergo a similar transformation, but at 523 K. In addition, many studies have focused on the electrochromic effect of this material because of its good electrochemical properties, making it a potential electrochromic counter electrode. Vanadium pentoxide is an n-type semiconductor with a band gap of approximately 2.3 eV, which determines its use as a good chemoresistive gas sensor [3,10–12]. The crystal structure of $V_2O_5$ contains orthorhombic cells with a lattice constant a = 1.151 nm. $V_2O_5$ shows highly anisotropic optical and electrical properties [13]. Vanadium pentoxide is used in a wide range of applications because of its thermal stability or wide band gap. The researchers' attention was also drawn to thermoelectrical properties, such as the Seebeck coefficient (S). The variability

of S with temperature tends to decrease with increasing temperature [14]. Because of the high temperature coefficient of resistance (TCR), it can be used in microbolometric devices. There are many different techniques that can be used to deposit $V_2O_5$ thin films: chemical vapor deposition, thermal deposition, laser ablation, sol-gel technology, and one of the most commonly used, i.e., magnetron sputtering [15–17]. Advantages of the magnetron sputtering method are low cost, high deposition rate, and uniform thickness. Magnetron sputtering also maintains reproducible properties and a large sputtering surface. Depending on the deposition technique and process conditions (such as process temperature or gas flow rate), vanadium pentoxide films may have different structural, electrical, and optical properties.

In recent years, many researchers have focused on the properties of annealed vanadium oxide thin films. For example, Akl et al. [18] investigated the influence of thermal annealing in a vacuum at 673 K of vanadium pentoxide on its microstructure and optical properties. The XRD spectra showed its polycrystalline nature, and it was observed that the peak intensity related to the (001) plane decreased after the annealing process, while the transmittance increased. Similar research results on the structural and optical properties of magnetron-sputtered vanadium oxides were also reported by Sieradzka et al. [19]. The authors showed that as deposited thin films were amorphous, while annealing at 673 K for 2 h caused phase transition to nanocrystalline $V_2O_5$ and simultaneous increase in the transmittance in the visible and near infrared wavelength range. The effect of thermal annealing on the electrical properties of $V_2O_5$ was also investigated, among others, by Mustafa Öksüzoğlu et al. [20], who studied the change of resistance values as a function of temperature. A slight broadening of the optical band gap occurred after annealing at 453 K. On the other hand, further annealing at 503 K caused a decrease in the resistance and the optical band gap, which was related to the phase change from amorphous to nanocrystalline structure. Moreover, a significant influence of temperature on electrical properties was observed by Manil Kang et al. [13], who found that the Seebeck coefficient increased with increasing temperature, but with a tendency to stabilize at a certain level. The gas sensing properties towards the hydrogen of $V_2O_5$ were investigated by J. Huotari et al. [21] at temperatures of 448 K and 563 K. On gas exposure, the resistivity of the thin films decreased. This phenomenon may be related to the water formation reactions, which add electrons to the surface, and thus the resistivity of n-type thin films decreased. However, for the measurements of the highest concentrations of $H_2$ at 448 K, a sudden change in the direction of response was observed, which was explained by the switching of the conduction mechanism. It was explained that the thin film may have both n-type and p-type conduction mechanisms, and the switch of the dominance of one of them depends on the surrounding conditions.

In view of the current state of knowledge, it is necessary to conduct thorough studies on the optical, electrical, and gas sensing properties of vanadium pentoxide. There is a need to make a comprehensive analysis of the structural properties that influence their optoelectronic properties (e.g., resistivity, type of electrical conduction, optical transmission, energy band gap) of $V_2O_5$ thin films, since such studies are scarce. Moreover, to date, it seems that $V_2O_5$ is not suitable for the development of gas sensing layers that work at operating temperatures above the metal-insulator transition temperature of 526 K [3]. Therefore, it is not suitable for work in harsh environments such as combustion or automotive exhaust. As stated by Alrammouz et al. [3], there is still a great challenge to further increase the quality of vanadium pentoxide to overcome its performance limitation. Therefore, in this work, detailed research on the effect of post-process annealing on the microstructure, morphology, optical, electrical and gas sensing properties of $V_2O_5$ thin films deposited by gas impulse magnetron sputtering was shown. The annealing temperature ranged between 423 K and 673 K. The conducted studies showed that proper annealing of $V_2O_5$ leads to changes in their optoelectronic and gas sensing properties.

## 2. Materials and Methods

Vanadium oxide thin films were deposited using the gas impulse magnetron sputtering method described in detail in [22]. Thin films were prepared with an Ar:$O_2$ gas ratio of 2:1. The deposition time was equal to 80 min, resulting in a thickness of 420 nm. Thin films were deposited on fused silica, silicon, and alumina substrates. Thin films deposited on silica were used to evaluate optical and structural properties, on silicon to determine the surface morphology and cross-section, and on alumina substrates for electrical and gas sensing measurements. Vanadium oxide thin films were annealed in an ambient air in a Nabertherm RS (80/300/11) tubular furnace at temperatures from 423 K to 673 K. The temperature increase rate was equal to 200 °C/h and thin films were held for 2 h at each annealing temperature. The furnace was then cooled down without any cooling media.

Structural properties were determined based on grazing incidence X-ray diffraction (XRD) measurements with a constant incidence angle of X-ray radiation of 3° using Empyrean PANalytical diffractometer (Panalytical, Malvern, UK) with CuKα X-ray source with a wavelength of 1.5406 Å. The average crystallite size was calculated employing Scherrer's equation [23]. The surface and cross-section morphology of vanadium oxide thin films were measured using the SEM/Xe-PFIB FEI Helios scanning electron microscope (Hilsboro, OR, USA).

Transmission spectra were measured using an Ocean Optics QE65000 spectrophotometer and a Mikropack DH-BAL 2000 light source (Ocean Optics, Largo, FL, USA) in the range from 250 to 1000 nm. Based on obtained results the average transparency in the visible wavelength range, cut-off wavelength, and optical band gap energy were calculated for as-deposited and annealed vanadium oxide thin films.

Current–voltage characteristics were measured using a Keithley SCS4200 characterization system and a M100 Cascade Microtech probe station (Keithley Instruments LLC, Cleveland, OH, USA) and were used to calculate sheet resistance and resistivity of as-deposited and annealed vanadium oxide thin films. Arrhenius plots were used to determine the activation energy based on temperature-dependent resistivity measurements performed in the temperature range of 300 K to 350 K.

To compare the optical and electrical properties of as-deposited and annealed vanadium oxide thin films the figure of merit (Q) coefficient was calculated using the following Equation (1):

$$Q = \frac{T^{10}}{\rho} \tag{1}$$

where *T*—average optical transmission in the visible wavelength range, $\rho$—resistivity at room temperature ($\Omega \cdot$cm).

Thermoelectric characteristics were measured using the FLUKE 8846A voltmeter (Fluke, Everett, WA, USA) and the Instek mK1000 temperature controller (GW Instek, Taipei, Taiwan). To determine the Seebeck coefficient (S), the temperature difference ($\Delta T$) between the two electrical contacts, i.e., "hot" and "cold", was established and increased to 50 K. Afterwards, the characteristic of the thermoelectrical voltage as a function of the temperature difference between two opposite contacts is determined and, on the basis of the slope, the Seebeck coefficient is calculated.

The gas sensing properties of the as-deposited and annealed vanadium oxides were measured for diluted hydrogen (3.5%) in argon. The change in resistance during gas sensing was determined by the Agilent 34970A data acquisition system (Agilent Technologies Inc., Santa Clara, CA, USA). Two experiments were carried out, in the first of which the operating temperature of the gas sensing process was maintained at 473 K, and vanadium oxides annealed at temperatures from 473 K to 673 K were measured. In the second experiment, vanadium oxide annealed at 673 K was measured at operating temperatures in the range of 473 K to 673 K.

## 3. Results

The XRD patterns of the deposited thin films are shown in Figure 1. It can be seen that as-deposited thin films were amorphous, which was caused by the low sputtering process temperature and relatively long target-substrate distance. Thus, the sputtered particles did not have enough energy to form a crystalline structure. However, thermal treatment at 423 K caused the transition from amorphous to nanocrystalline structure. The crystallite size calculated for the most intense peak related to the (110) crystal plane remained the same for thin films annealed up to 523 K and was equal to ca. 23 nm. Further annealing caused its increase to ca. 27 nm (Figure 1b). A similar effect was observed by Sieradzka et al. [19], who showed recrystallization of vanadium oxide after post-process annealing. Furthermore, Liu Y. et al. [24] carried out research in which the average crystallite size increased with increasing annealing temperature.

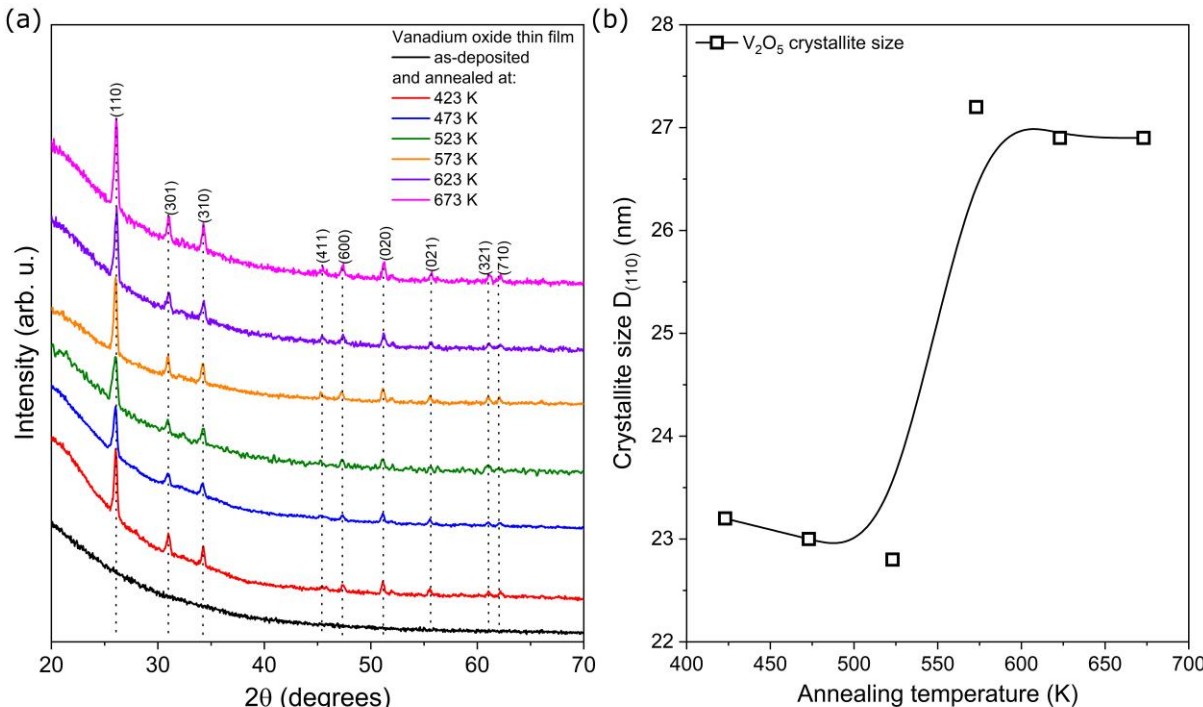

**Figure 1.** X-ray diffraction measurements results: (**a**) diffraction patterns and (**b**) an average crystallite size of $V_2O_5$ thin films annealed at various temperatures calculated for (110) crystal plane.

Images of optical microscope investigations performed in the transmission mode are shown in Figure 2. It is clearly seen that the as-deposited vanadium oxide is crack-free and homogenous with only some dust visible at the surface. However, annealing just at 423 K caused a significant change, and the observed images showed that the thin films were made from micrometer-sized cracks and grains, which had a substantial effect on their transparency. That is, the as-deposited thin films had a yellow colour and were clear, but after annealing at the lowest temperature, they changed their character to milky and matt.

The SEM images in Figure 3 show the surface morphology and cross-section of thin films annealed at various temperatures. The as-deposited vanadium oxide thin film was characterized by a smooth and plain cross-section, which is in agreement with XRD results showing amorphous behaviour. However, it can be seen that post-process annealing at 423 K resulted in the phase transition from amorphous to nanocrystalline $V_2O_5$ that also caused the appearance of small grains. An increased annealing temperature caused an increase in surface roughness and an increase in grain size. Furthermore, the grains grew abnormally, their sizes were inhomogeneous, and their shape was irregular. In addition, annealing at 623 K and 673 K resulted in the formation of the coarse grains that grow from the substrate to the surface of the thin film. It is worth noting that the increase in surface

roughness and the formation of grains with voids between them causes an increase in the porosity leading to an enhanced surface-active area of the thin films. This is a crucial phenomenon in the context of the gas sensing properties.

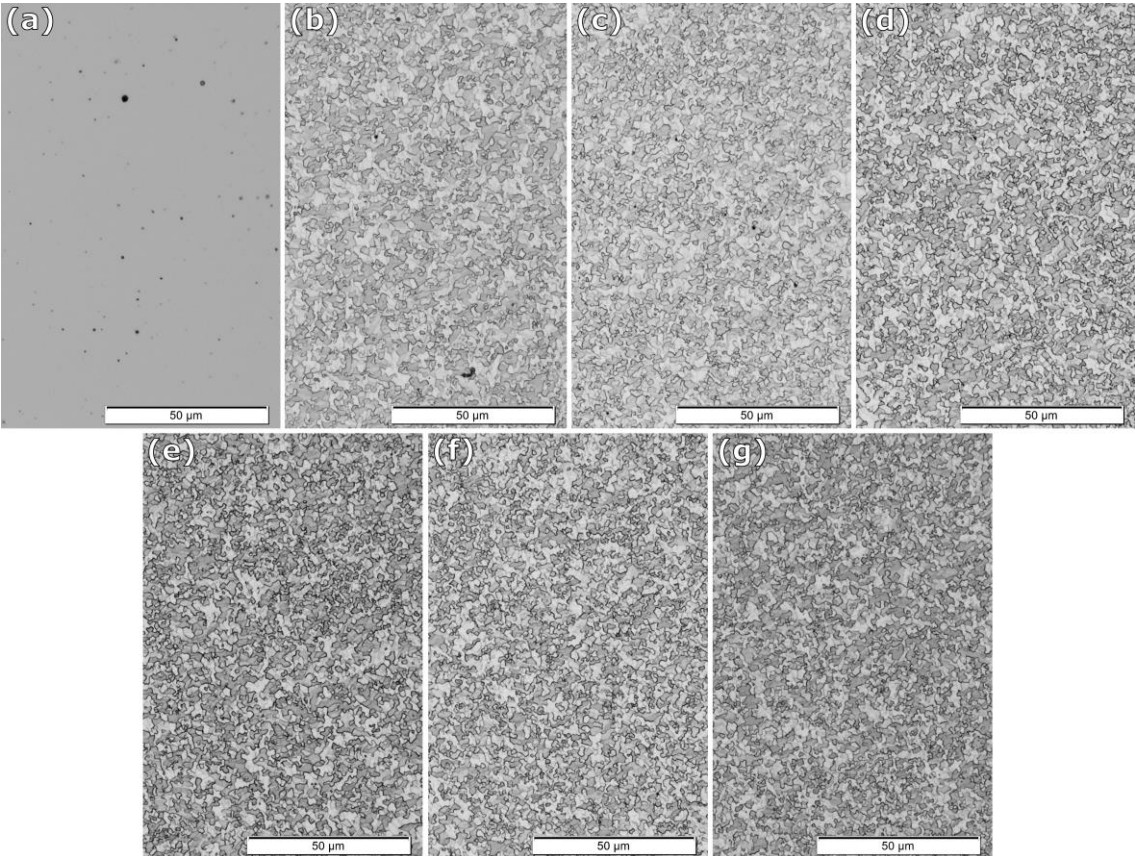

**Figure 2.** Optical microscopy images of the surface of $V_2O_5$ thin films annealed at various temperatures: (**a**) as deposited, (**b**) 423 K, (**c**) 473 K, (**d**) 523 K, (**e**) 573 K, (**f**) 623 K and (**g**) 673 K.

Transmission spectra of the $V_2O_5$ thin films annealed at various temperatures are shown in Figure 4a, while the average transmission in the visible wavelength range calculated as an integral below each curve is presented in Figure 4b. It can be seen that as-deposited thin films are quite transparent above 450 nm, while annealing caused a significant decrease in the transmission coefficient. With an increase in the annealing temperature up to 523 K, the transmission coefficient is gradually decreasing in the wavelength range above 500 nm. It might be related to the occurrence of large cracks visible at the optical microscope images (Figure 2), which possibly cause light scattering as their dimensions are similar or larger to the light wavelength. Moreover, one can see the decrease in the cut-off wavelength towards shorter wavelengths with an increase in the annealing temperature, which could be considered as a blueshift. However, it can be again caused by the occurrence of cracks. Thin films annealed at 523 K and above had very similar transmission characteristics. As shown in Figure 4b, the average transmission in the visible wavelength range is approximately 50% for as-deposited thin films and it decreases significantly to approx. 30–33% with annealing. Meng et al. [25], Aiempanakit et al. [26] and Zhu et al. [7], who showed that transmittance decreased with increasing annealing temperature, which was caused by increased surface roughness and light scattering.

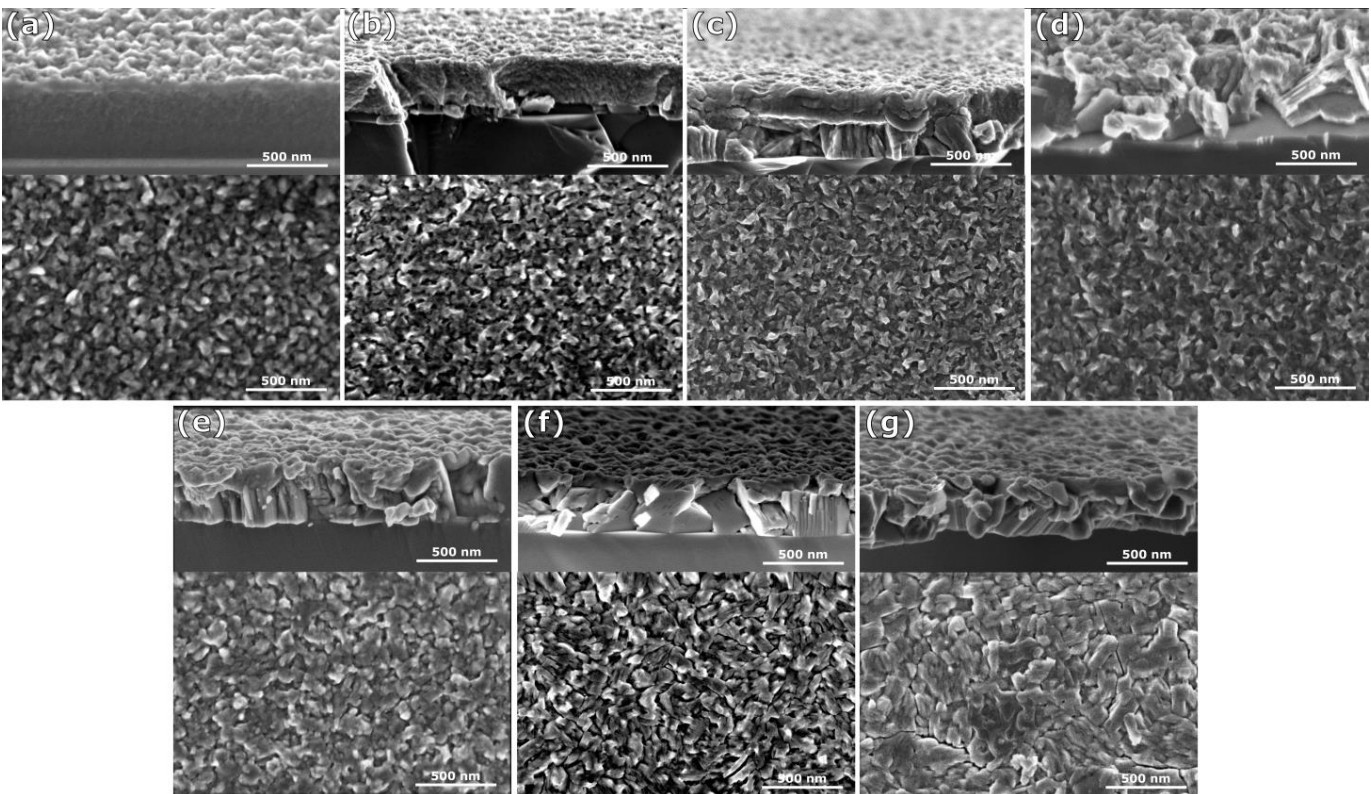

**Figure 3.** SEM images of the surface and cross-section morphology of $V_2O_5$ thin films annealed at various temperatures: (**a**) as deposited, (**b**) 423 K, (**c**) 473 K, (**d**) 523 K, (**e**) 573 K, (**f**) 623 K and (**g**) 673 K.

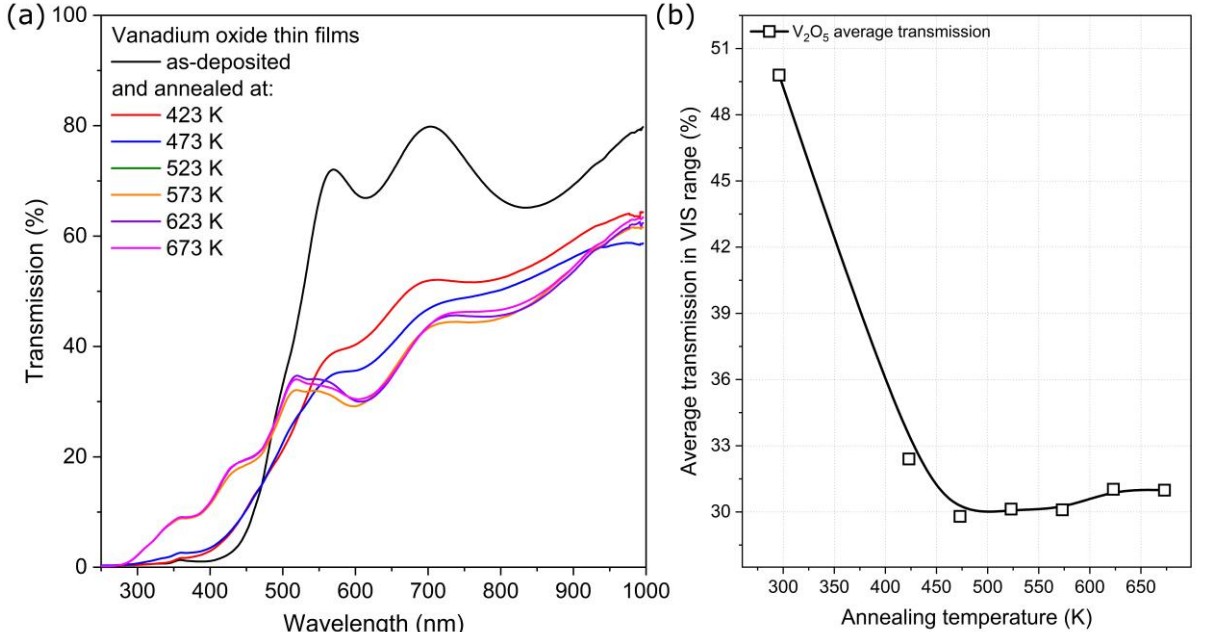

**Figure 4.** Results of: (**a**) transmission spectra of $V_2O_5$ thin films annealed at various temperatures and (**b**) the average transmission of vanadium oxides in the VIS wavelength range.

The indirect band gap energy for the thin vanadium oxide films was calculated using Tauc plots [6,27] by extrapolating the linear portion of the curves from the plot of $(\alpha h\nu)^{1/2}$ in the function of the photon energy ($h\nu$). All plots are shown in Figure 5 with marked

lines used for the determination of $E_g^{opt}$, while comparison of the fundamental absorption edge and optical band gap energy of $V_2O_5$ thin films annealed at various temperatures are shown in Figure 6. The optical band gap energy for as-deposited thin films is equal to 2.08 eV, while annealing at 423 K and 473 K caused its decrease to 1.71 and 1.64 eV, respectively. This contradicts the conclusions that can be drawn by determining the cut-off wavelength ($\lambda_{cut-off}$). In this case, annealing resulted in a $\lambda_{cut-off}$ decrease, which is opposite to the results of the determination of $E_g^{opt}$. In this case, however, it is necessary to take into account the change in the slope of the transmission curve, which is smaller than in the case of the as-deposited thin film, which also causes a change in the slope of the Tauc plot and, consequently, the calculation of the value of optical band gap energy. Annealing at temperatures of 523 K and above causes a significant decrease in the cut-off wavelength to ca. 280 nm and an increase in the $E_g^{opt}$ to approx. 2.7 eV, but it is worth noting that both values are already constant despite the increase in the annealing temperature. It is worth highlighting that the values of the optical energy band gap of 2.6–2.7 eV are in agreement with the literature [28].

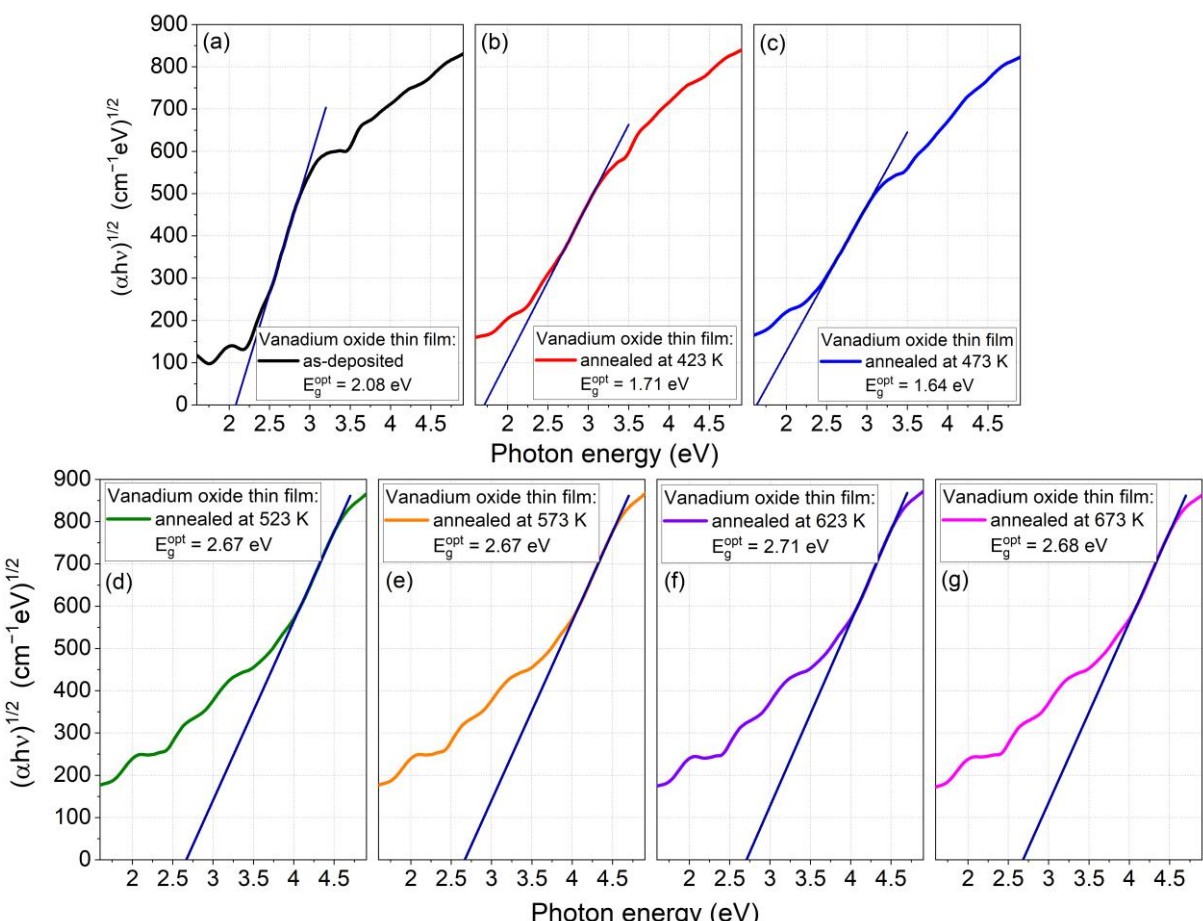

**Figure 5.** The determination of the optical band gap energy of $V_2O_5$ thin films annealed at various temperatures: (**a**) as deposited, (**b**) 423 K, (**c**) 473 K, (**d**) 523 K, (**e**) 573 K, (**f**) 623 K and (**g**) 673 K.

Current–voltage characteristics of the as-deposited and annealed vanadium oxide thin films are presented in Figure 7a. All results revealed a linear dependence between current and voltage. Taking into consideration the slope of each current–voltage curve, the resistance of the thin films can be determined. The conductivity as a function of temperature is shown in Figure 7b, and it increases with an increase in temperature, which is typical for intrinsic semiconductors [24]. The highest conductivity was obtained for the sample annealed at 473 K. Furthermore, the conductivity of the thin films annealed at 623 K and 673 K decreased significantly. These variations might be corelated with the

change in surface morphology shown in SEM images. Similar phenomena were observed by M. Öksüzoğlu et al. [20], who examined the influence of post-annealing temperature on resistivity of the DC sputtered vanadium oxides. It was shown that there is a temperature over which the morphological changes can cause an increase in the resistivity.

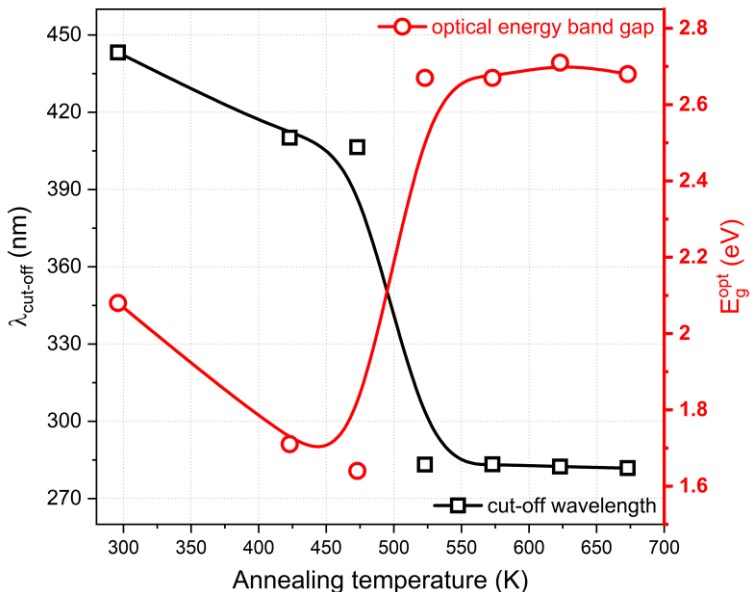

**Figure 6.** Comparison of the fundamental absorption edge and optical band gap energy of $V_2O_5$ thin films annealed at various temperatures.

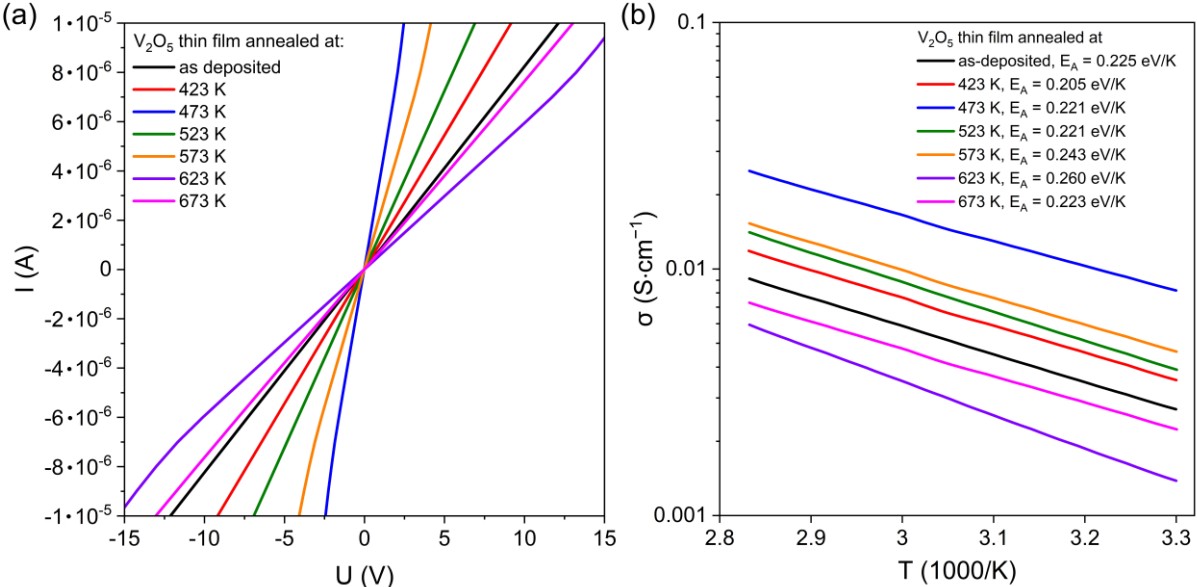

**Figure 7.** Results of electrical properties investigations: (**a**) current-voltage characteristics and (**b**) conductivity as a function of 1000/T measured for vanadium oxide thin films annealed at various temperatures.

The activation energy ($E_a$) was calculated for each sample using the Arrhenius law Equation (2) and linearly fitted slopes of the conductivity as a function of the temperature, using the following equation:

$$\sigma = \sigma_0 exp\left(\frac{E_A}{kT}\right) \tag{2}$$

where $k$ is the Boltzmann constant and $T$ is the absolute temperature.

Variations in $E_A$ values for various annealing temperatures were rather negligible, ranging between 0.205 and 0.260 eV/K. Very similar results were also obtained by M. Öksüzoğlu et al. [20] for $V_2O_5$ thin films measured at temperatures of 323 K to 623 K, which indicated the thermal activation of charge carriers into conduction band.

The dependence of the resistivity of the vanadium oxide thin films on their annealing temperature is shown in Figure 8a. At first, the resistivity is decreasing, most probably due to the decreasing number of defects, and at annealing temperatures above 473 K it starts to increase. However, it is worth noting that in each case the change is not significant.

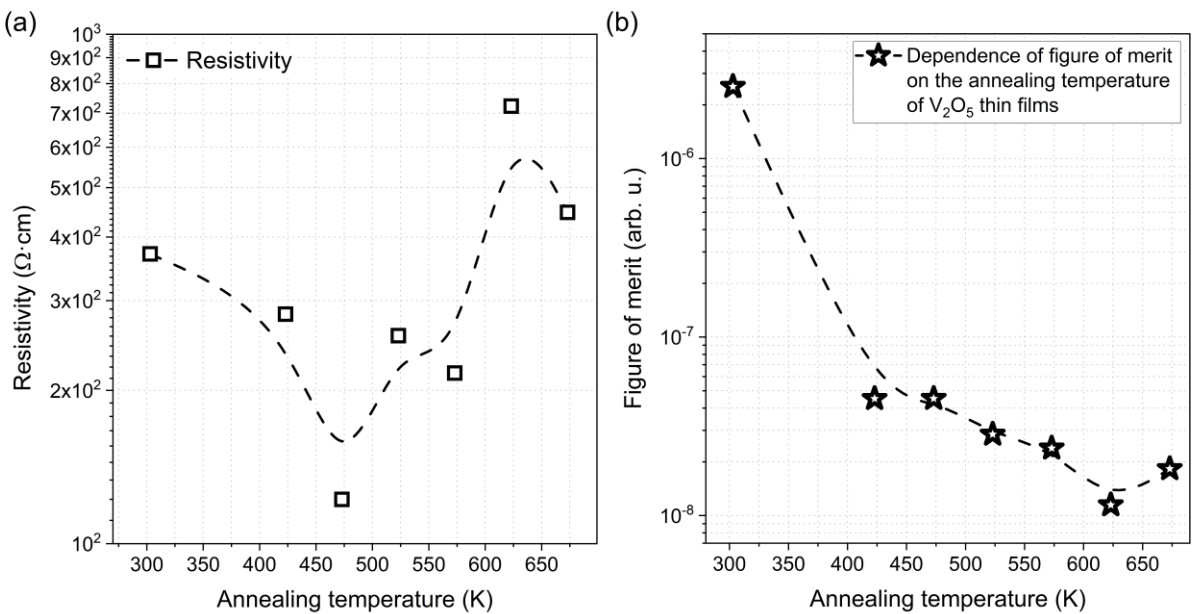

**Figure 8.** Dependence of: (**a**) resistivity and (**b**) figure of merit of $V_2O_5$ thin films on the post-process annealing temperature.

Knowing the average transparency in the visible wavelength range and resistivity of $V_2O_5$ thin films annealed at various temperatures, it was possible to compare their optoelectronic performance with the use of the figure of merit ($Q$) according to Equation (1). It was found that the value of the figure of merit was decreasing with the increase in the annealing temperature, as shown in Figure 8b. Since the value of resistivity is at the same order for each thin film, the figure of merit was decreasing mainly due to the decrease in the average transmission. Therefore, the highest $Q$ was obtained for as-deposited vanadium oxide, which exhibited the highest average transmission and just intermediate resistivity.

Thermoelectrical measurements were used to determine the value and sign of the Seebeck coefficient (S) of as-deposited and annealed $V_2O_5$ thin films. The thermoelectrical voltage changes for each sample as a function of the temperature difference between two electrical contacts are presented in Figure 9. The negative sign of the Seebeck coefficients indicates the electron type of electrical conductivity, while the positive sign indicates the hole-type conductivity [29]. The results showed that each investigated thin film was an n-type semiconductor and the S was in the range from ca. 20 to 92 μV/K, while the highest Seebeck coefficient was obtained for $V_2O_5$ film annealed at 523 K.

Gas sensing properties in a 3.5% hydrogen environment of as-deposited and annealed vanadium oxide thin films were also examined. Changes in resistance as a function of time are shown in Figure 10. The resistive chemical sensors are based on the phenomenon of band energy bending. This process might be a result of many phenomena; however, the main cause is gas adsorption, which is correlated with oxygen vacancies created during temperature treatment. On these reactive radicals, the oxygen is physiosorbed, when exposed in air at room temperature. Thermal treatment causes electrical charge carriers from the deeper part of the thin film to be trapped by adsorbed $O_2$ molecules. Exposition

of a sensing thin film to a proper gas atmosphere causes changes of the number of trapped charge carriers, and as a result, the resistance of the sensing film is changing as well.

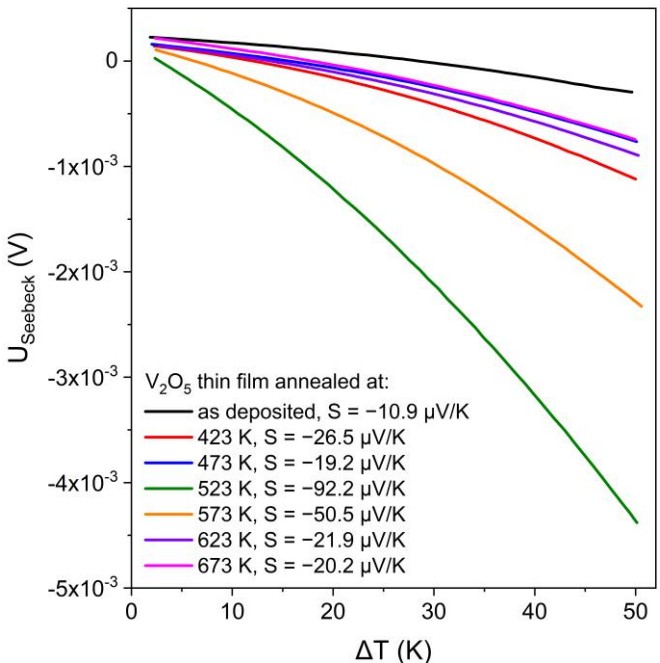

**Figure 9.** Thermoelectrical voltage characteristics of as-deposited and annealed vanadium oxide thin films with calculated Seebeck coefficients.

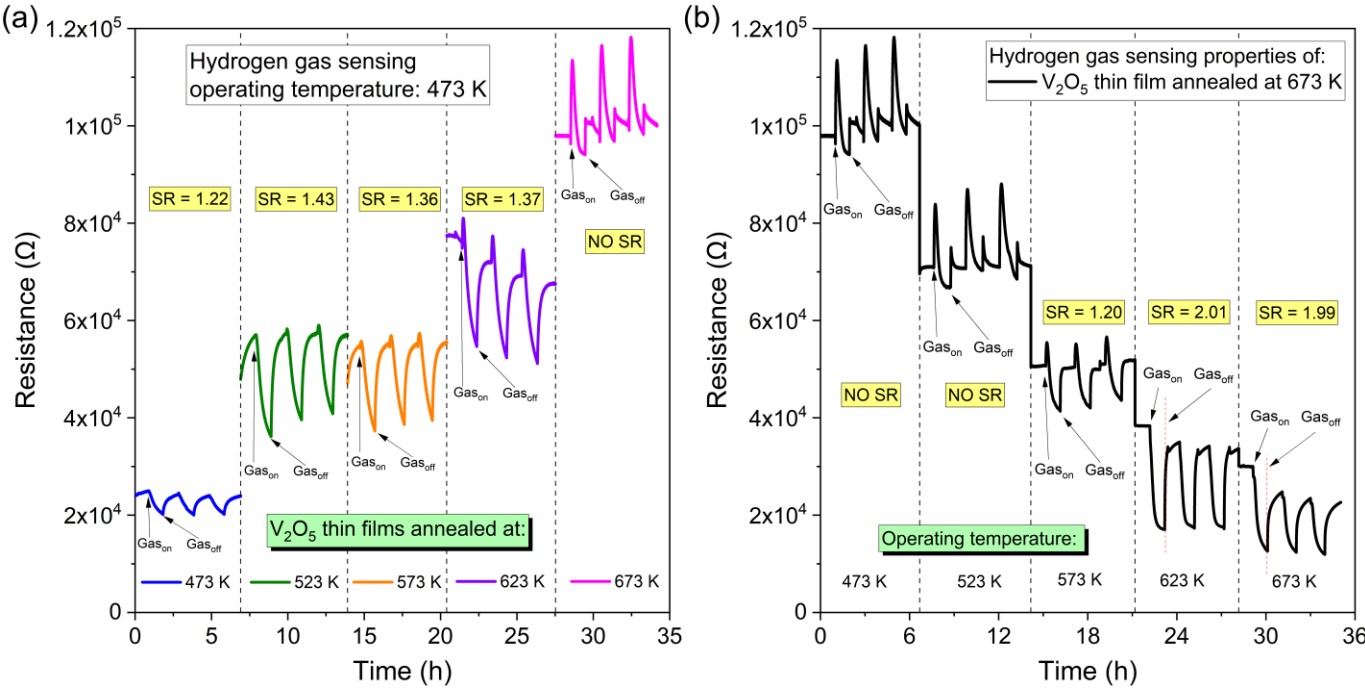

**Figure 10.** Electrical resistance changes upon exposure to diluted hydrogen (3.5% in Ar) of $V_2O_5$ thin films: (**a**) annealed at various temperatures and measured at operating temperature of 473 K and (**b**) annealed at 673 K and measured at various operating temperatures.

The sensor response (*SR*) was investigated for thin films annealed at temperatures between 473 K and 673 K. The sensor response is one of the parameters specifying the

gas sensor and is defined as the ratio of resistance measured in air ($R_{air}$) and resistance measured in hydrogen ($R_{gas}$):

$$SR = \frac{R_{air}}{R_{gas}}$$

Gas sensing measurements were performed in two scenarios: (1) at a constant operating temperature of 473 K for each annealed thin film and (2) for thin films annealed at 673 K at various operating temperatures. The results of the investigation are shown in Figure 10a with marked *SR* for each measurement. It was found that at a constant operating temperature of 473 K the best sensor response of 1.43 was obtained for the thin film annealed at 523 K, which also exhibited the highest Seebeck coefficient. For thin films annealed at a temperature of 473 K, the *SR* was only 1.22, while for films annealed at higher temperatures it was ca. 1.36. For the sample annealed at 673 K, the *SR* was not possible to determine.

To determine how the operating temperature at which the measurement is performed affects the sensor response, measurements of the $V_2O_5$ thin film annealed at 673 K were carried out at operating temperatures ranging from 473 K to 673 K. Results are shown in Figure 10b. In this case, the best *SR* of 2.01 and 1.99 was obtained for the sample measured at 623 K and 673 K, respectively. Considerably lower *SR* was obtained for measurements carried out at 573 K and was equal to 1.20, while at 473 K and 523 K the *SR* was impossible to determine. It is assumed that the appearance of resistance peaks visible for $V_2O_5$ thin film annealed at 673 K at lower operating temperatures may be caused by the volumetric response preceded by the surface sensor response.

Better *SR* for annealed thin films at higher temperatures and measured at higher operating temperatures might be corelated with changes of surface properties, i.e., increase in roughness and porosity of the thin films. According to the XRD results, the annealing temperature above 423 K had a negligible effect on the microstructure as a result of just a slight increase in the crystallite size. Therefore, it can be assumed that the surface of the thin films played a crucial role in the gas detection properties.

## 4. Conclusions

Amorphous vanadium oxide thin films were deposited using gas impulse magnetron sputtering without using additional substrate heating or electrical bias. Post-process annealing caused a change in structural, surface, optical, and electrical properties. Annealing at a relatively low temperature of just 423 K caused a phase transition from amorphous to the nanocrystalline $V_2O_5$ with a crystallite size of approximately 23 nm, while above 523 K a further increase in crystallite size to 27 nm was observed. Moreover, thermal treatment caused formation of cracks, micrometer-sized grains, and increase in the surface roughness. The effect of post-process annealing also had a significant influence on the transparency, which decreased form ca. 50% for as-deposited vanadium oxide to ca. 30%–33% for annealed films. Measurements of electrical properties revealed that thermal treatment did not cause a substantial change in the resistivity of the $V_2O_5$ thin films, which remained at the order of $10^2$ $\Omega \cdot$cm. Taking into consideration results of the transparency in the visible wavelength range and resistivity, a figure of merit was calculated to evaluate the performance of optoelectronic properties of $V_2O_5$ thin films. It was found that as a result of the highest transmission level for as-deposited thin films, the best figure of merit was also obtained for this film. Furthermore, measurements of Seebeck coefficient were carried out and revealed electron type of conduction for each vanadium pentoxide thin film. Investigations of gas sensing properties performed for diluted hydrogen showed that in the case of measurements at quite low operating temperature of 473 K the best sensor response was obtained for the $V_2O_5$ thin film annealed at 523 K. Whereas measurements carried out for the $V_2O_5$ thin film annealed at 673 K exhibited the best sensor response for operating temperature of 623 K. The results obtained are promising and show the possibility of using vanadium oxide films prepared by magnetron sputtering as promising thin-film materials for gas sensor applications. Detailed studies regarding changes of the oxidation state and

roughness of the surface are foreseen to be carried out in the future works, which will definitely give additional insight into the gas sensing mechanism of annealed vanadium pentoxide thin films.

**Author Contributions:** Conceptualization, M.M.; methodology, M.M., S.K. and J.D.; validation, M.M.; formal analysis, M.M., S.K. and J.D.; investigation, M.M. and S.K.; resources, M.M.; data curation, M.M.; writing—original draft preparation, M.M., S.K. and J.D.; writing—review and editing, M.M.; visualization, M.M.; supervision, M.M.; project administration, M.M.; funding acquisition, M.M. All authors have read and agreed to the published version of the manuscript.

**Funding:** This work was co-financed from the sources given by the Polish National Science Centre (NCN) as a research project number 2020/39/ D/ST5/00424 in the years 2021–2024.

**Institutional Review Board Statement:** Not applicable.

**Informed Consent Statement:** Not applicable.

**Data Availability Statement:** Not applicable.

**Conflicts of Interest:** The authors declare no conflict of interest.

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
