# Peer review of "Effect of Annealing on the Microstructure, Opto-Electronic and Hydrogen Sensing Properties of V2O5 Thin Films Deposited by Magnetron Sputtering"

_coatings, doi:10.3390/coatings12121885_

Round 1

Reviewer 1 Report

This manuscript employed the gas impulse magnetron sputtering method to deposit V2O5 thin films. The crystal structure, opto-electronic, and sensing properties were studied after annealing. The systematic characterization expanded from materials to sensor performance. However, to improve the current manuscript, I believe the following comments should be addressed:

Comments 1): The introduction provides the application of V2O5, however, the motivation of this work is not elaborated. What problem is this work going to solve? I recommend the authors add a paragraph at line 82 to convince the public of the necessity of this work.

Comments 2): For the Tauc plots in Figure 5, the baselines are missing, which cast doubt on the accuracy of the extrapolation and the determination of the band gap. Please show the baselines.

Comments 3): The authors choose 423 K as the lowest annealing temperature. Crystallization of V2O5 is proved by XRD. Why does the annealing temperature start at 423 K? What happens below 423 K?

Comments 4): The details of annealing need to be provided (lines 96). What is the pressure, ambient, and temperature increase rate? This may help the public to understand the origin of “oxygen vacancies created during temperature treatment”. (line 276)

Comments 5): The authors attribute the change in the V2O5 properties to the crystallinity and cracks after stepwise annealing treatment. Why the Seebeck coefficient and the conductivity do not change monotonously with the annealing temperature?

Comments 6): References need to be added at the beginning of the introduction. (line 26-28)

Comments 7): Academic writing has to be precise. In line 123, The author mentions “ the gradient was increased from 0 to 50 K.” How is 0 K achievable? What method did the author use to achieve 0 K? A typo is found by the reviewer. In line 101, the word “using” is duplicated. To be accepted by Coatings, any similar error needs to be addressed.

Author Response

Answers to the report of Reviewer #1

on the manuscript entitled: Effect of annealing on the microstructure, opto-electronic and sensing properties of V2O5 thin films deposited by gas impulse magnetron sputtering

Authors: Michał Mazur, Szymon Kiełczawa, Jarosław Domaradzki

Authors:

We would like to express our gratitude for your remarks, which let us improve our manuscript. We have taken them into account in the revised version of our paper.

Answering to the Reviewer’s remarks, we have introduced some revisions in the manuscript.

Reviewer:

The introduction provides the application of V2O5, however, the motivation of this work is not elaborated. What problem is this work going to solve? I recommend the authors add a paragraph at line 82 to convince the public of the necessity of this work.

Authors:

According to the Reviewer’s comment the article was extended at the end of the introduction as follows:

“In view of the current state of knowledge, it is necessary to conduct thorough studies on the optical, electrical, and gas sensing properties of vanadium pentoxide. There is a need to make a comprehensive analysis of the structural properties that influence their optoelectronic properties (e.g. resistivity, type of electrical conduction, optical transmission, energy band gap) of V2O5 thin films, since such studies are scarce. Moreover, to date, it seems that V2O5 is not suitable for the development of gas sensing layers that work at operating temperatures above the metal-insulator transition temperature of 526 K [3]. Therefore, it is not suitable for work in harsh environments such as combustion or automotive exhaust. As stated by Alrammouz et al. [3], there is still a great challenge to further increase the quality of vanadium pentoxide to overcome its performance limitation. Therefore, in this work, a detailed research on the effect of post-process annealing on the microstructure, morphology, optical, electrical and gas sensing properties of V2O5 thin films deposited by gas impulse magnetron sputtering was shown. The annealing temperature ranged between 423 K and 673 K. The conducted studies showed that proper annealing of V2O5 leads to changes in their optoelectronic and gas sensing properties.”

Reviewer:

For the Tauc plots in Figure 5, the baselines are missing, which cast doubt on the accuracy of the extrapolation and the determination of the band gap. Please show the baselines.

Authors:

The baseline was added. The determination of the energy band gap was done using the interpolation of the Tauc plot and calculated taking into consideration the slope of the linear fit. The accuracy is about 0.02 eV.

Fig. 5. The determination of the optical band gap energy of V2O5 thin films annealed at various temperatures

Reviewer:

The authors choose 423 K as the lowest annealing temperature. Crystallization of V2O5 is proved by XRD. Why does the annealing temperature start at 423 K? What happens below 423 K?

Authors:

At lower temperature of 373 K the crystallization of V2O5 does not occur, therefore a temperature of the 423 K was taken as the lowest annealing temperature. There is also no significant change of the optical and electrical properties at lower temperatures.

Reviewer:

The details of annealing need to be provided (lines 96). What is the pressure, ambient, and temperature increase rate? This may help the public to understand the origin of “oxygen vacancies created during temperature treatment”. (line 276)

Authors:

According to the Reviewer’s comment following details of the annealing were provided to the manuscript:

“Vanadium oxide thin films were annealed in an ambient air in a Nabertherm RS (80/300/11) tubular furnace at temperatures from 423 K to 673 K. The temperature increase rate was equal to 200°C/h and thin films were held for 2 hours at each annealing temperature. The furnace was then cooled down without any cooling media.”

Reviewer:

The authors attribute the change in the V2O5 properties to the crystallinity and cracks after stepwise annealing treatment. Why the Seebeck coefficient and the conductivity do not change monotonously with the annealing temperature?

Authors:

This phenomenon needs further studies that Authors are doing right now. It seems that the Seebeck coefficient is somehow correlated with the crystallite size, i.e. the highest S is determined for thin films annealed at 523 K that exhibited the lowest crystallite size (D). Above this temperature the D value increased by ca. 15-20%. In turn, the lowest value of resistivity is obtained for thin films annealed at 473 K and above this temperature the resistivity increases (however, its value is still of the same order). This seems to be correlated with an increase of the grain size seen at SEM images where the grains become coarse for the annealing temperature of 523 K and above.

However, as stated above, further research is needed to distinguish which factor has the greatest influence on the electrical properties and is currently being undertaken.

Reviewer:

References need to be added at the beginning of the introduction. (line 26-28)

Authors:

Article was extended with some references:

“Metal oxides with electrical and optical properties suitable for the use in opto-electronics and sensor technology have been studied for many years [1]. The current interest in metal oxides is due to its properties, which are determined by oxygen vacancies [2]. One of the chemical compounds which is recently of great interest is vanadium oxide [3-5].”

[1] Granqvist, C.G. Electrochromics for smart windows: Oxide-based thin films and devices, Thin Solid Films 2014, 564, 1–38. https://doi.org/10.1016/j.tsf.2014.02.002

[2] Migas, D. B., Shaposhnikov, V. L., Rodin, V. N., & Borisenko, V. E. Tungsten oxides. I. Effects of oxygen vacancies and doping on electronic and optical properties of different phases of WO3. Journal of Applied Physics 2010, 108(9), 093713. https://doi.org/10.1063/1.3505688

[3] Alrammouz, R.; Lazerges, M.; Pironon, J.; Taher, I.B.; Randi, A.; Halfaya, Y.; Gautier, S. V2O5 gas sensors: A review, Sensor. Actuat. A-Phys. 2021, 332, 113179. https://doi.org/10.1016/j.sna.2021.113179

[4] Mjejri, I.; Rougier, A.; Gaudon, M. Low-cost and facile synthesis of the vanadium oxides V2O3, VO2, and V2O5 and their magnetic, thermochromic and electrochromic properties, Inorg. Chem., 2017, 56(3), 1734–1741. https://doi.org/10.1021/acs.inorgchem.6b02880

[5] Mounasamy, V.; Mani, G.K.; Madanagurusamy, S. Vanadium oxide nanostructures for chemiresistive gas and vapour sensing: a review on state of the art, Microchim. Acta 2020, 187(4), 253. https://doi.org/10.1007/s00604-020-4182-2

Reviewer:

Academic writing has to be precise. In line 123, The author mentions “ the gradient was increased from 0 to 50 K.” How is 0 K achievable? What method did the author use to achieve 0 K?

Authors:

Authors are grateful for the comment. In fact, the “gradient” is not the best word to describe the Seebeck coefficient measurements, while the “difference between the hot and cold electrical contacts” should be used instead. Therefore, the experimental part was changed accordingly:

“Thermoelectric characteristics were measured using the FLUKE 8846A voltmeter and the Instek mK1000 temperature controller. To determine the Seebeck coefficient (S), the temperature difference (ΔT) between the two electrical contacts, i.e. “hot” and “cold”, was established and increased to 50 K. Afterwards, the characteristic of the thermoelectrical voltage as a function of the temperature difference between two opposite contacts is determined and, on the basis of the slope, the Seebeck coefficient is calculated.”

Reviewer:

A typo is found by the reviewer. In line 101, the word “using” is duplicated. To be accepted by Coatings, any similar error needs to be addressed.

Authors:

Authors checked the manuscript for other errors and typos and corrected them.

Reviewer 2 Report

see the attachment

Author Response

Answers to the report of Reviewer #2

on the manuscript entitled: Effect of annealing on the microstructure, opto-electronic and sensing properties of V2O5 thin films deposited by gas impulse magnetron sputtering

Authors: Michał Mazur, Szymon Kiełczawa, Jarosław Domaradzki

Authors:

We would like to express our gratitude for your remarks, which let us improve our manuscript. We have taken them into account in the revised version of our paper.

Answering to the Reviewer’s remarks, we have introduced some revisions in the manuscript.

Reviewer:

Revision of the title,  „and sensing properties” (hydrogen sensing properties); „deposited by gas impulse magnetron sputtering” (deposited by magnetron sputtering)/ (deposited using high power impulse magnetron sputtering)/ (deposited by gas impulse magnetron sputtering) -  if this form is kept, a clear description of the method and the specific differences, including references in the introduction, is recommended.

Authors:

According to the Reviewer’s comment the title was changed as follows:

“Effect of annealing on the microstructure, opto-electronic and hydrogen sensing properties of V2O5 thin films deposited by magnetron sputtering

Reviewer:

From the hydrogen sensitivity measurements, it appears that the values are very low - is this explained by the fact that the gas concentration is low (3.5% below the explosion limit)? Would the sensitivity be influenced if the mixture were a usual one (hydrogen in the air)? Would the behavior of the semiconductor type change?

Authors:

The concentration of the hydrogen is low, just below the explosion limit of 4%. Moreover, it is diluted in argon, which may hinder the sensor response as compared to the hydrogen diluted in air. Especially, if the mixture of hydrogen and air has some humidity, then the sensor response should be a lot higher. The behaviour of the semiconductor type should not change as it is independent from the annealing temperature, i.e. all samples are n-type semiconductors and show a decrease of the resistance under the influence of hydrogen gas. Such behaviour is typical for n-type semiconductors. In this paper we focused our attention to analyse the changes of the optical and electrical properties and gas sensor response to hydrogen with the annealing temperature. We found that there is a specific annealing temperature and operating temperature of the gas sensor in which the sensor response is the highest among all samples.

Reviewer:

Wcm; S*cm-1  (W×cm; S×cm-1);

Authors:

Units were changed to W×cm and S×cm-1.

Reviewer:

It is recommended that the references be shorter and more concrete. To reflect more clearly the values of some parameters obtained from the optical and electrical properties, the type of semiconductor, hydrogen sensitivity values, etc.

Authors:

Authors tried to compare their results with the state of the art and literature reports. However, sometimes it hard since other Authors used different techniques to prepare their vanadium oxide thin films or nanostructures. While in the case of the gas sensing properties the medium in which hydrogen is diluted is also very important because it influence the sensor response.

Moreover, the introduction part had to be extended with the references according to the second Reviewer comment.

In the case of structural properties we included following references to the literature:

“A similar effect was observed by Sieradzka et al. [19], who showed recrystallization of vanadium oxide after post-process annealing. Furthermore, Liu Y. et al. [24] carried out research in which the average crystallite size increased with increasing annealing temperature.”

In the case of optical properties:

“As shown in Fig. 4b the average transmission in the visible wavelength range is approximately 50% for as-deposited thin films and it decreases significantly to approx.. 30-33% with annealing. Meng et al. [25], Aiempanakit et al. [26] and Zhu et al. [7], who showed that transmittance decreased with increasing annealing temperature, which was caused by increased surface roughness and light scattering.”

“The indirect band gap energy for the thin vanadium oxide films was calculated using Tauc plots [6, 27] by extrapolating the linear portion of the curves from the plot of (αhν)1/2 in the function of the photon energy (hν).”

“It is worth highlighting that the values of the optical energy band gap of 2.6-2.7 eV are in agreement with the literature [28].”

In the case of electrical properties:

“Similar phenomena were observed by M. ÖksüzoÄŸlu et al. [20], who examined the influence of post-annealing temperature on resistivity of the DC sputtered vanadium oxides.”

“Variations in EA values for various annealing temperatures were rather negligible, ranging between 0.205 and 0.260 eV/K. Very similar results were also obtained by M. ÖksüzoÄŸlu et al. [20] for V2O5 thin films measured at temperatures of 323 K to 623 K, which indicated the thermal activation of charge carriers into conduction band.”

“The negative sign of the Seebeck coefficients indicates the electron type of electrical conductivity, while the positive sign indicates the hole-type conductivity [29].”

Round 2

Reviewer 1 Report

The manuscript has been significantly improved after revision. I suggest accepting the current version for publication.